# Multi-omics integration accurately predicts cellular state in unexplored conditions for *Escherichia coli*

Minseung Kim[1,2], Navneet Rai[2,*], Violeta Zorraquino[2,*] & Ilias Tagkopoulos[1,2]

A significant obstacle in training predictive cell models is the lack of integrated data sources. We develop semi-supervised normalization pipelines and perform experimental characterization (growth, transcriptional, proteome) to create Ecomics, a consistent, quality-controlled multi-omics compendium for *Escherichia coli* with cohesive meta-data information. We then use this resource to train a multi-scale model that integrates four omics layers to predict genome-wide concentrations and growth dynamics. The genetic and environmental ontology reconstructed from the omics data is substantially different and complementary to the genetic and chemical ontologies. The integration of different layers confers an incremental increase in the prediction performance, as does the information about the known gene regulatory and protein-protein interactions. The predictive performance of the model ranges from 0.54 to 0.87 for the various omics layers, which far exceeds various baselines. This work provides an integrative framework of omics-driven predictive modelling that is broadly applicable to guide biological discovery.

[1] Department of Computer Science, University of California, Davis, California 95616, USA. [2] Genome Center, University of California, Davis, California 95616, USA. * These authors contributed equally to this work. Correspondence and requests for materials should be addressed to I.T. (email: itagkopoulos@ucdavis.edu).

Traditionally, host-specific data integration has been small in scale and limited to two layers[1–5], mostly because of the lack of data across multiple layers for the same experimental conditions[6,7]. More recently, we have witnessed omics resources that cover organism-specific gene expression data, one such effort being the COLOMBOS database that combines multi-layer, multi-organism data with manually curated condition information[1]. As we accumulate more data within and across layers, such vertical and horizontal integration becomes more efficient and meaningful. Integration over more than two layers leads to lower false discovery rates and an enhanced picture of various cellular mechanisms and adaptive responses[8,9]. It is also critical for data-driven modelling, which until now has relied on custom, more restricted omics data sets[2–5]. Despite the fact that repositories of raw data have existed for more than a decade[7], the development of databases with two or more omics layers is in an early stage[6]. Recently, the MOPED database was created to address this issue, with a multi-omics resource portal that combines 250 publicly available protein and mRNA abundance profiles of four organisms (human, mouse, worm and yeast)[10]. Other efforts such as KBase are complementary and aspire to provide various bioinformatics services at all levels ranging from alignment and assembly of raw sequencing data, phylogenetic analysis, protein annotation and other modelling tools[11]. The scientific community has already acknowledged these efforts as well as the lack of a database with normalized multi-layered data across experimental conditions, with sufficient meta-data and quality control[8,9,12,13].

There are many challenges when it comes to multi-omics compendium construction. First, systematic biases exist due to technological platforms, laboratories and analysis methods[14,15]. Experiments have primarily focused on sampling one layer of biological organization, hence making it difficult to have multi-layer data for the same condition. Indeed, for *Escherichia coli* we have only 33 samples with the trifecta of transcriptome, proteome and metabolome, and even in those cases, not simultaneously. In addition, many data sets are mis-annotated or lack meta-data, a fact that requires close inspection of the published work and communication with the authors. Concomitantly, the sheer size of training data needed to avoid model overfitting and the dimensionality of the experimental space are equally daunting, which limited the generalization performance of past modelling approaches[5,16–19]. These discrepancies create obstacles for the application of machine learning and modelling techniques, which aim to learn from data[9,13–15]. Subtle normalization issues can also have a substantial impact to the quality and utility as a training set of any compendium[20]. For example, since the total RNA per cell fluctuates, the standard assumption that total RNA/cell doesn't change, and thus expression distributions are identical across varying conditions is prone to produce false discoveries in the downstream analysis[21]. This observation is especially important for developing a data compendium where experiments correspond to several conditions (media, abiotic factors, and so on) which all together affect global factors such as growth rate[22] and the total RNA/cell[18].

Yet, normalized multi-omics compendia are essential for the genome-wide models used in a number of fields[23]. Among the many mechanistic cell models that simulate biophysical cell properties, E-Cell[24] was one of the first cell models that simulated cell behaviours of 127 genes based on a series of differential equations. A notable recent development in this area was the whole-cell model of *Mycoplasma genitalium* that integrates genome-scale biophysical properties of all ranges of molecules in an organism for specifically simulating a cell division process[25]. Similarly, the *E. coli* ME-model predicts fluxes and gene expression levels given nutrient availability and genetic perturbations by formulating it into an optimization problem of metabolic reactions[26]. In the realm of data-driven models, the transcriptional model of *Halobacterium salinarum* predicts modular responses in the transcriptome layer given the expression levels of 72 transcription factors and 9 environmental cues[27]. More recently, PROM integrates flux balance analysis (FBA) and probabilistic inference from large-scale transcriptome data[28], while expression balance analysis (EBA) adds over FBA the data-driven approach to predict genome-wide transcriptional levels by constructing a large-scale transcriptome compendium (2,242 profiles)[5].

In this work, we describe two advances in the field of multi-omics integration and multi-scale modelling. First, we present Ecomics, a normalized, well-annotated, multi-omics database for *E. coli*, developed to provide high-quality data and associated meta-data for performing predictive analysis and training data-driven algorithms. This compendium houses 4389 normalized expression profiles across 649 different conditions. Moreover, we present the Multi-Omics Model and Analytics (MOMA) platform, an integrated model that learns from the Ecomics and other available network data to predict genome-wide expression and growth, which shows higher performance than several baselines and two recent metabolic-expression models.

## Results

**The multi-omics compendium and the integrative modelling**. To build a multi-omics compendium for *E. coli*, we first aggregated data from available literature and public databases (Fig. 1). We then created a normalization pipeline that is based on semi-supervised approaches to remove systematic biases due to data collection, platform differences, batch effects, conditions and analysis methods (Fig. 2)[21,22]. We identified data with missing or potentially mis-annotated entries and supplement meta-data information both through communication with the respective laboratories and through systematic experimental characterization in our lab. Currently (February 2016), Ecomics has 4389 genome-wide profiles over 649 conditions, coming from 65 *E. coli* K-12 strains (Supplementary Fig. 1), 286 genetic perturbations (for example, knock-outs), 112 media (Supplementary Fig. 2) and 52 stresses.

Second, the MOMA platform is an integrated model that learns from the Ecomics and other available network data to predict genome-wide expression and growth (Fig. 3 and Supplementary Fig. 3). Given an input of 612 features that encompass genetic (strain, genetic perturbation) and environmental (medium, stress) factors, the proposed model predicts the genome-scale expression and concentration of genes, proteins and metabolites (5453 molecular species), metabolic fluxes (2,382) and growth rates. The predictive performance of the model compares favourably against several baselines and in comparison to two recent metabolic-expression models.

**A highly biased sampling of the experimental space**. Figure 4a is a snapshot of the *E. coli* K-12 omics universe. The lack of omics data for any single condition is striking: from the 649 conditions represented, only 11 conditions have two or more omics layers (6.19% of the profiles) and only one condition has all three layers (0.41% of the profiles). The distribution of profiles to conditions is heavily skewed (Fig. 4b,c and Supplementary Figs 4,5A,6,7). Profile replicates are generally reproducible and highly variable depending on the specific strain (CV = 0.28 ± 0.09; Supplementary Fig. 5A), medium (CV = 0.26 ± 0.14; Supplementary Fig. 6) and stress (CV = 0.24 ± 0.11; Supplementary Fig. 7).

**Targeted experimentation in Ecomics**. The gene ontology (GO) coverage in Ecomics is unexpectedly low: only 7.2% of molecular functions, 48.4% of biological processes and 31.4% of KEGG pathways are represented in the compendium. Although the challenges associated with gene ontologies are many and known[29], GO term coverage provides a proxy on how diverse the information within the compendium is. To maximize the GO coverage of Ecomics, we first investigated which conditions are likely to disrupt unrepresented biological processes and functions. We assumed that a GO term is represented in the compendium if the compendium has profiles where one or more genes which include that GO term have been perturbed. Based on the resulting ranked list, we performed transcriptional profiling (RNA-Seq, see Methods) for the top 16 genetic knockouts (in triplicate; 48 profiles total). We also transcriptionally profiled and included in Ecomics nine KO experiments that we had identified before as being highly informative for both GO coverage and model performance[5] (Supplementary Table 1). After the inclusion of these new transcriptional profiling results, the coverage increases by a 63.9, 24.3 and 19.8% to molecular function, biological process and KEGG pathway representation, respectively (Fig. 4f, Supplementary Fig. 8, Supplementary Data 1, Supplementary Methods).

**Reconstructing an omics-based ontology**. We evaluated the similarity of the various strains, media and stresses, based on their effect in genome-scale expression (Supplementary Methods). Interestingly, the omics-derived strain ontology does not follow the sequence-based reconstruction, with a cophentic correlation of 0.21 (Fig. 4d; Supplementary Fig. 9). This result challenges the prevailing notion that distance in the genomic space translates into cellular state similarity[30] and argues that small genetic changes can lead to large differences in the genome-wide molecular state. Similarly, the ontology reconstruction for omics-based medium and stress provide a data-driven perspective that directly correlates with how the organism perceives and responds to environmental changes (Fig. 4e; Supplementary Figs 10C,11).

Investigation of the genes and processes that led to the stress ontology reconstruction provides the genetic basis of their commonalities (Fig. 4e). We analysed the differential expression in pairs of stresses clustered together with using all other stresses as controls (Supplementary Data 2). In the case of isoleucine-limitation and octanoic acid, previous work has shown the notable difference in amino acid isotoper enrichments after octanoic acid exposure in *E. coli*[31], and has also suggested carboxylic acids may negatively impact the function and/or integrity of the cell membrane. Interestingly, the top unique genes in this stress pair include many genes involved in the outer membrane including: *ompX, slyB, acrA* and *tolC*[32], while the link between bacterial membrane synthesis and amino acid starvation has been previously established[33]. In the case of nanoparticle and butanol stress, *cpxR* shows the highest statistical significance and is known to be one of the two most differentially regulated genes during n-butanol stress in *E. coli*[34] as well being involved in bacteria-nanoparticle interactions. In the cluster of hypoxia and Indole-3-acetic acid stress, the genes *hybF, aceK, metN, talB* are involved in anaerobic respiration[35,36] and energy/central metabolism where Indole-3-acetic acid is known to affect their relevant pathways in *E. coli*[37]. For cold and heat shock, the ribosome is similarly responsive in both stresses[38] and this, together with temperature sensitive responses is what drives the high similarity in these two stresses. The top differentially expressed genes that are common in this pair are *rplK, rplL, rplV, rplO, rplC, rplJ, rplM,* all of which are ribosome related[32]. For ATP and NADH limitation, both ATP and NADH are energy molecules with levels directly influenced by external energy sources. The top differentially expressed genes in that pair, include dppB, a component of the *DppABCDF* dipeptide transport system[39] and *gadE* which is regulated through a cAMP receptor[40], have been shown to be altered by glucose. In the case of osmotic and acidic stresses the link is unclear, as their most informative differentially expressed genes, the dehydrogenase *glpB* and the cell division protein *ftsQ* are implicated in other processes, although our results are in agreement with previous

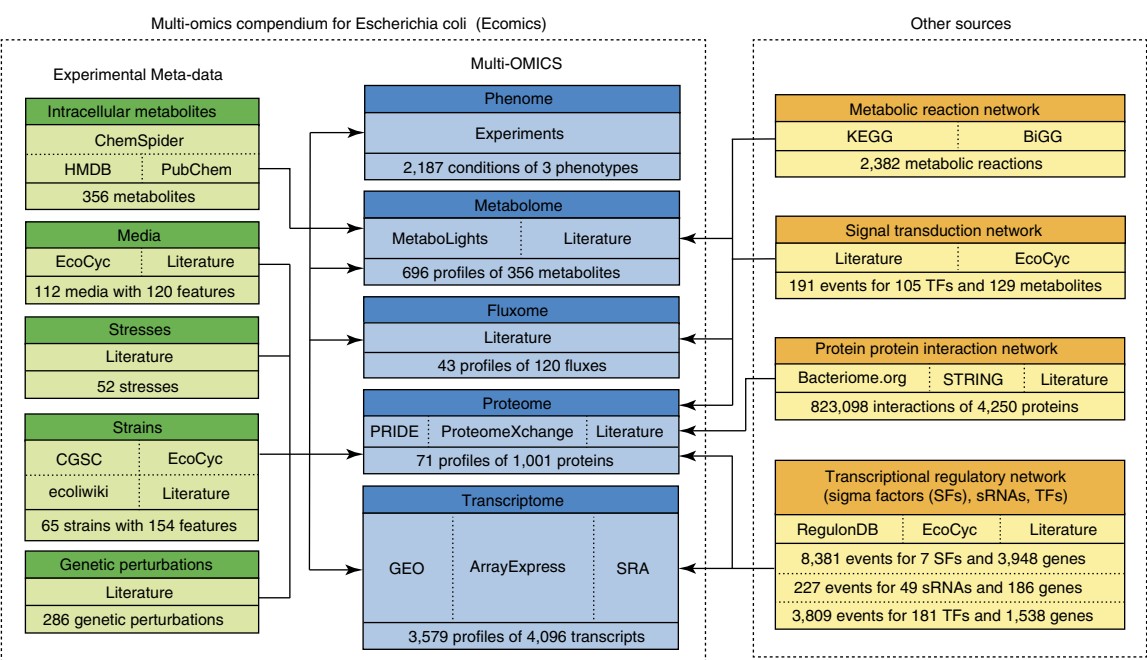

**Figure 1 | Overview of the Ecomics data sources.** The Ecomics compendium is divided into two major parts, the multi-omics genome-scale profiles and the experiment meta-data. In addition to the Ecomics compendium, we have curated genome-scale interaction data for signal transduction, transcriptional, protein and metabolic layers.

reports observing their altered expression under these stress conditions[41].

**Factors that affect expression variability in *E. coli*.** Expression variability is strain, media and stress dependent (Supplementary Figs 5B,10A-B,12, Supplementary Methods) beyond biases

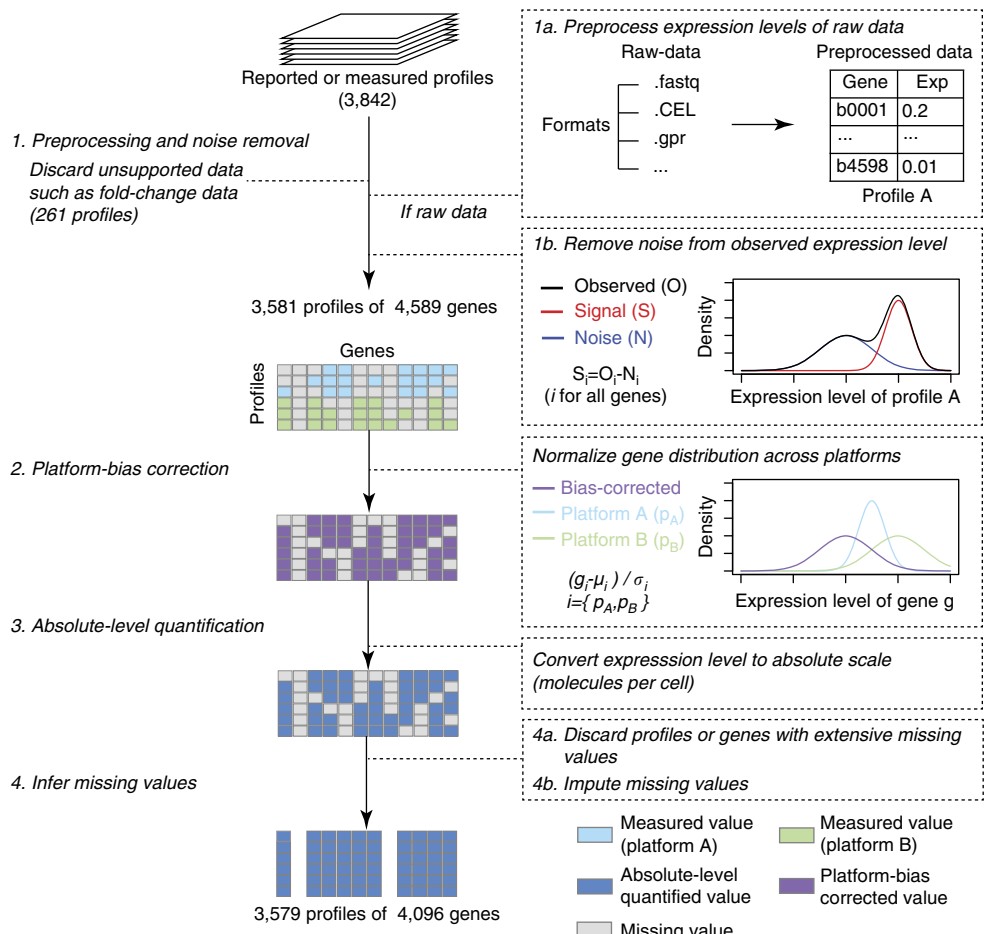

**Figure 2 | Overview of Ecomics data processing steps.** Raw profiles were first processed to calculate expression levels or molecular concentrations. Noise removal and bias correction normalized data across different platforms. Absolute level quantification converted relative expression level to absolute numbers for cross-condition comparison. Final quality control removed low quality entries and inferred missing information.

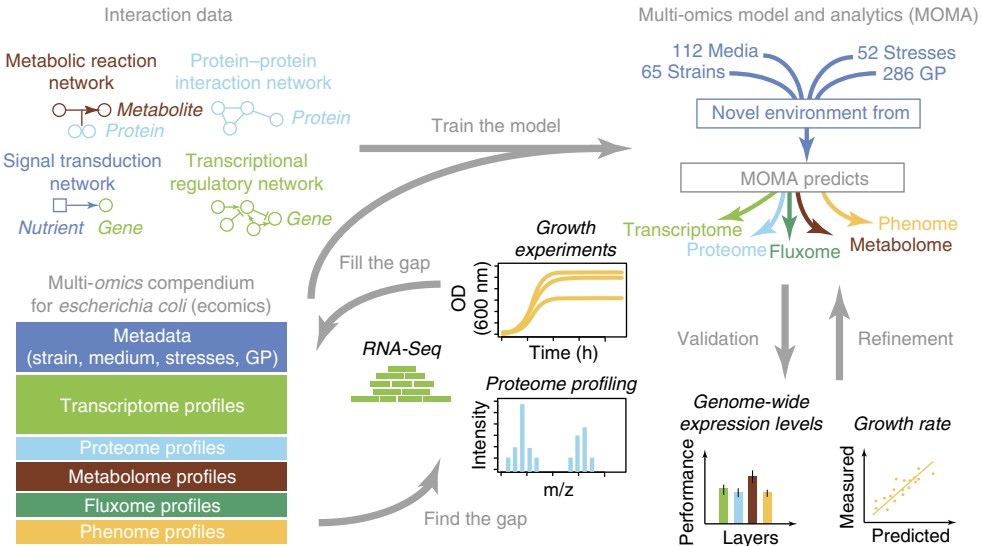

**Figure 3 | Overview of the multi-omics framework.** A multi-layered predictive model constructed from interaction data (metabolic reaction network, protein-protein interaction network, signal transduction network, transcriptional regulatory network) and trained on the multi-omics compendium. The model is composed of the transcriptome, proteome, metabolome, fluxome and phenome layers. GP: genetic perturbations.

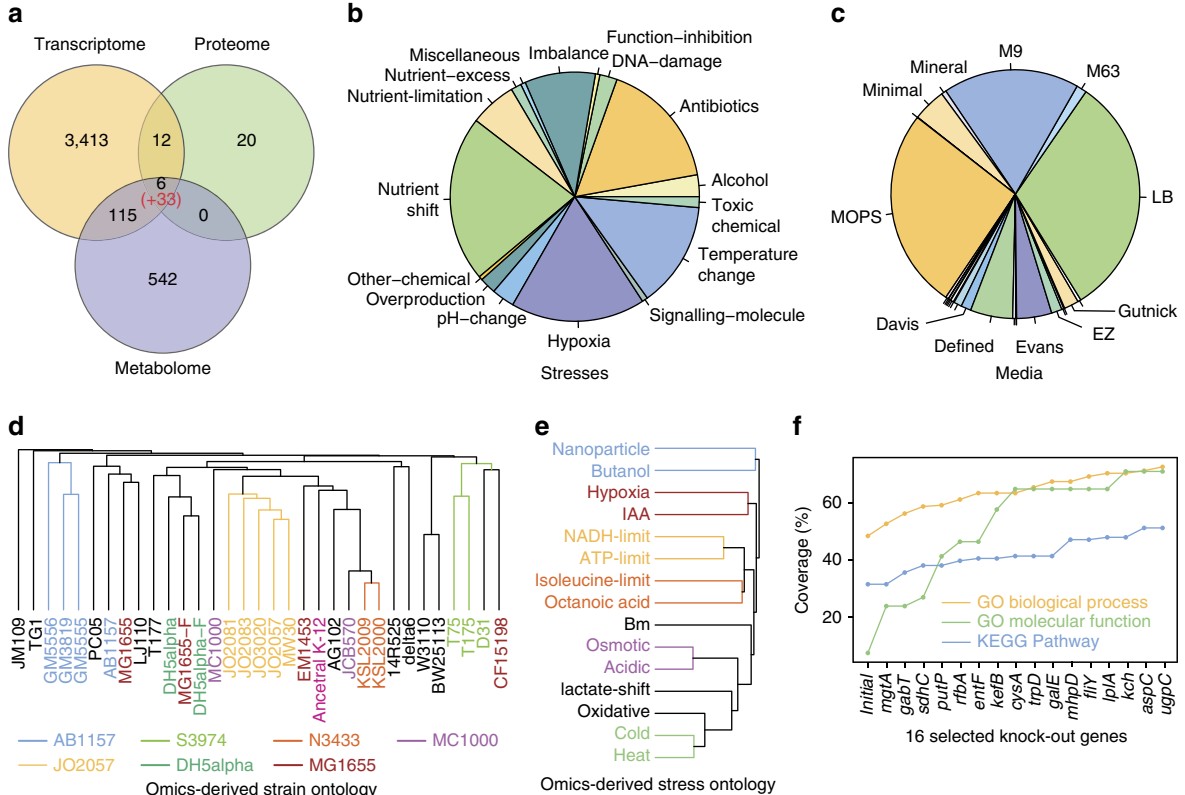

**Figure 4 | Analysis of the Ecomics compendium. (a)** Number of profiles across the transcriptome, proteome and metabolome layers. The number in parentheses represents the profiles targeted for core metabolism[62]. (**b**) Stress distribution in Ecomics. The 52 stresses used in the compendium are grouped into 16 categories. Imbalance refers to the imbalance between production and consumption of reactive species (for example, oxidative, nitrosative). Alcohol represents exposures in alcoholic compounds such as butanol or ethanol. Overproduction refers to the overexpression of macromolecules such as fatty-acid or recombinant protein. Function-inhibition refers to the inhibition of cellular function such as lipid synthesis or peptidoglycan. (**c**) Media distribution in Ecomics. The 112 media used in the compendium are grouped into 25 basic categories (14 displayed; K, modified complex, BHI, ACSH, TSB, TB, SynH, TPM2, dYT, phosphate, DMA, DMEM, GMM, GGM are not displayed). (**d**) Strain ontology based on omics profiles. Only transcriptional profiles in LB medium were used. Colour represents strain derivatives. (**e**) Stress ontology based on omics profiles. Only transcriptional profiles in M9/MOPS media and without stress or genetic perturbation were used. Colour represents common biological processes that are enriched in each stress pair: signal transduction (cyan); central metabolism (red), transport (yellow); cell division (purple); ribosome formation (green). (**f**) GO coverage enrichment. 16 gene knock-out experiments were iteratively selected based on the number of unrepresented GO terms that were perturbed. Coverage on the y axis is the percentage of functional terms represented in the compendium. The 'initial' point on the x axis indicates the Ecomics GO coverage before enrichment.

resulting from experimental settings or sample size (Pearson Correlation Coefficient (PCC) is between 0.01 and 0.03 in both cases, for laboratory and sample size bias). To identify key genetic markers that affect expression variance, we performed LASSO regression on the genotypic map of 154 features and expression variability for the 25 strains, which led to the identification of 6 genetic features. This approach was applied to media, identifying 11 nutrient features. A detailed analysis of these results that includes the genes identified and the mechanisms involved in variability of strains, media and stresses is available in Supplementary Methods.

**MOMA—an integrative model for predicting molecular state.** We developed the MOMA framework that is based on a recurrent neural network (RNN)[42] with regularization of sigmoid functions for the transcriptome layer, LASSO regression and ensemble learning[43] for the proteome and metabolome layers, and weighted summation for their integration (Fig. 5a; Supplementary Figs 4,13–15; Supplementary Methods). MOMA was designed to be trained based on the multi-omics and interaction data that are in the Ecomics compendium and predict multi-omics expression for novel conditions (given as an input vector of the strain, medium, stress and genetic perturbation; see Methods).

Because the predictive ability relies on the data used to train the model, each attribute in the new condition to predict should exist in at least one entry of the training set (in this case, Ecomics). Since there are 65 strains, 112 media, 52 stresses and 286 genetic perturbations in Ecomics, the theoretical space it can cover is given by their product, or about 285 million conditions.

In our previous work, we used regression on transcription factors to predict gene expression (EBA model described in ref. 5). While this work is still applicable here, it is limited in its ability to parameterize the environmental inputs and does not account for feedback loops that are ubiquitous in gene regulatory networks. To address these issues, the RNN architecture was used with an input feature vector corresponding to the genetic and environmental background and gene expression as the output. The network architecture and optimal parameters were selected through cross-validation techniques from the training data (Supplementary Fig. 16). For the proteome and metabolome layer, we evaluated several techniques on their ability to integrate and learn from the training data (more in the Methods section and Supplementary Methods). In every single layer, MOMA significantly outperformed the baseline predictions (based on random, mean and wild-type profiles) for expression and molecular concentration, even when only the information of that

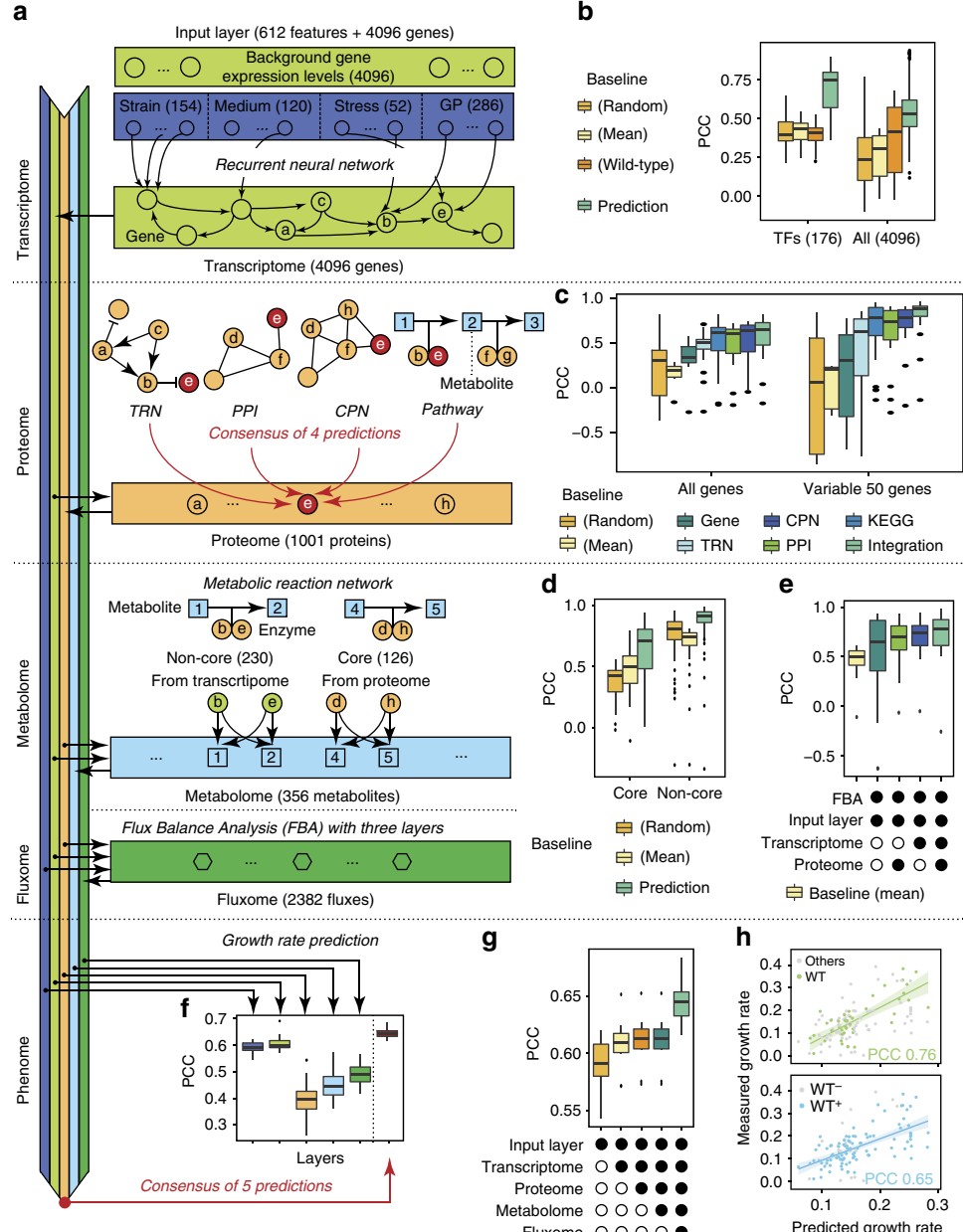

**Figure 5 | Model architecture and prediction performance.** (**a**) The data and work flow over the five modules, one for each layer. (**b**) Prediction performance of the transcriptome module for all 178 TFs and all genes. For TFs (PCC $0.68 \pm 0.14$, left panel), $P < 10^{-13}$ for random baseline (PCC $0.41 \pm 0.09$), $P < 10^{-13}$ for mean baseline (PCC $0.41 \pm 0.07$), $P < 10^{-14}$ for wild-type baseline (PCC $0.40 \pm 0.07$). For all genes (PCC of $0.54 \pm 0.15$, right panel), $P < 10^{-15}$ for random baseline (PCC $0.25 \pm 0.17$), $P < 10^{-15}$ for mean baseline (PCC $0.26 \pm 0.14$), $P < 10^{-15}$ for wild-type baseline (PCC $0.36 \pm 0.22$) (**c**) The prediction performance for the proteome module. The number of proteins for each module (all proteins; 50 most variable): TRN (250;20), KEGG (547;36), PPI (1000;50), CPN (847;50), Integration (1001;50). Baselines from linear protein inference from transcriptional level (PCC $0.34 \pm 0.18$, $P < 10^{-2}$), random (PCC $0.24 \pm 0.34$, $P < 10^{-2}$) and mean (PCC $0.16 \pm 0.10$, $P < 10^{-4}$) is shown. (**d**) Metabolite prediction (PCC $0.65 \pm 0.21$; 126 metabolites) from 75 protein expression levels in core metabolism (comparison with baselines: random, PCC $0.36 \pm 0.13$, $P < 10^{-6}$; mean, PCC $0.47 \pm 0.19$, $P < 10^{-3}$). Metabolite prediction (PCC $0.87 \pm 0.15$; 230 metabolites) in non-core metabolism from 4096 gene expression levels (comparison with baselines: random, PCC $0.77 \pm 0.17$, $P < 10^{-13}$; mean, PCC $0.70 \pm 0.13$, $P < 10^{-15}$). (**e**) Flux prediction, evaluated over 32 different conditions. Performance (PCC of $0.72 \pm 0.24$) was compared to that of FBA without other layers (PCC of $0.65 \pm 0.39$, $P < 10^{-1}$), without transcriptome (PCC of $0.67 \pm 0.21$, $P < 10^{-1}$), without proteome (PCC of $0.70 \pm 0.20$, $P < 10^{-1}$) and mean baseline (PCC of $0.50 \pm 0.11$, $P < 10^{-8}$). (**f**) Performance of growth rate prediction from each layer (input, transcriptome, proteome, metabolome, fluxome from left) and the consensus of predictions from five layers (right). Performance to random baseline is not shown (PCC $= -0.01$; $P < 10^{-5}$) (**g**) Additive effect of additional layers in growth rate prediction. (**h**) Comparison of predicted growth rate and measured growth rates for 60 novel wild-type conditions (top) and for 101 conditions with wild-type information present in the training set (bottom). In all performance evaluation, three baselines are used: the random baseline represents prediction from mean expression level of randomly selected profiles in the training set; the mean baseline is the prediction from mean expression level of all profiles in the training set; the wild-type baseline is the prediction by mean expression levels of wild-type profiles in the training set. In all cases, leave-one-condition-out (LOCO) cross-validation was used. The error bar indicates standard error of the mean. Wilcoxon rank sum test was used for all statistical tests.

given layer was used for prediction (Fig. 5b-g). Furthermore, the performance of the model is substantially higher (PCC of 0.58 versus 0.19 with baselines 0.28 and 0.05, respectively) when comparing the training set of the Ecomics normalization (based on an absolute scale) to that of COLOMBOS (based on fold-change) on otherwise identical raw data sets (Supplementary Fig. 17).

For the transcriptome layer, the prediction of the expression of the 176 transcription factors (TFs) is significantly better when compared to the prediction of the expression of all 4096 genes (PCC: $0.76 \pm 0.07$ versus $0.54 \pm 0.15$, Fig. 5b). The robustness of prediction varies across combinations of strain, media and stresses with the observed variation being in agreement with the variability analysis, as described in the previous section (Fig. 5b and Supplementary Fig. 18). For example, our predictions are more accurate in defined medium among 19 media (PCC $= 0.84 \pm 0.3$ versus $0.51 \pm 0.14$; Supplementary Fig. 18B). We found that the model performance is uniform across the different pathways (Supplementary Fig. 19). Furthermore, 56% (48.8%) of GO biological processes, 62.8% (10.3%) of GO molecular functions, 97.5% (87.5%) of GO cellular process and 58.6% (41.3%) of KEGG pathways have a PCC higher than the mean of 0.54 (number in parentheses is percentage of cases that also have a statistically significant higher PCC than the wild-type baseline; Supplementary Data 3). In addition, the model performance was significantly enhanced by the amount of data in the compendium. A tenfold increase in data size adds a 15% increase in performance (measured by PCC) in the general case (Supplementary Table 2).

The expression level for each of the 1001 proteins was predicted through an ensemble method that integrates information from four sources (Supplementary Methods, Supplementary Fig. 14): the transcriptional regulatory network (TRN), the protein-protein interaction (PPI) network, the co-expressed protein network (CPN) and other pathway information. We evaluated each of these methods individually, as well as their integration through an Ensemble method where the protein expression level is the mean of the predicted expression level from each of the four prediction modules. The evaluation was performed in an Ecomics-derived data set of 18 profiles (5 conditions) with expression levels in both the transcriptional (4096 transcripts) and proteome layers (1001 proteins). As shown in Fig. 5c, the integration of four methods shows higher coverage and higher performance when compared with the four individual methods. In terms of protein coverage, the integration can predict all 1001 proteins, comparable only to the PPI method (1000 proteins), while the other three individual methods can predict a substantially lower number of proteins (250 for TRN, 547 for KEGG, 847 for CPN). However, the prediction performance of the PPI method (PCC: $0.48 \pm 0.26$ for PPI) is lower than using their integration (PCC: $0.55 \pm 0.26$). The integration method also outperforms the other two individual methods (PCC: $0.41 \pm 0.23$ for TRN; $0.47 \pm 0.23$ for KEGG) and is close to that of the CPN method (PCC: $0.52 \pm 0.24$), although the latter has a 15.4% less coverage. To directly compare sets of proteins that are covered in all methods, we focused their performance in the top 50 most variable proteins that are common among the five sets. Our results show that the integration (ensemble method) outperforms all other combinations with a PCC of $0.77 \pm 0.27$, which is substantially better than all others (CPN, PCC $= 0.69 \pm 0.27$; PPI, PCC $= 0.60 \pm 0.36$; KEGG, PCC $= 0.66 \pm 0.30$, TRN, PCC $= 0.37 \pm 0.49$). Interestingly, predicting first the target gene expression from the mRNA expression levels of the corresponding genes does not perform well, achieving a PCC of $0.34 \pm 0.18$ in the general case and PCC of $0.18 \pm 0.51$ for the 50 most variable proteins (Supplementary Fig. 20).

Previous studies report that genome-wide expression levels between transcripts and proteins in cell populations of *E. coli* show some but not strong correlation[44–46]. More specifically, an $R^2$ of 0.47 between average mRNA and protein concentration was reported in ref. 45. This has suggested that the mRNA expression level of a gene is not the best proxy for estimating the concentration of the corresponding protein. Our results show that the concentration of a protein can be highly predicted ($0.79 \pm 0.21$, $R^2$ of log-scaled concentration for 1001 proteins) when we collectively use expression levels of the genes that are functionally related to the protein where four different network sources are used to find functional relationships of genes. This result argues that integrating genome-wide interaction and expression data is beneficial for predicting protein levels.

For predicting the metabolome layer, we first investigated which layer (transcriptome, proteome) can provide the necessary information to build a better predictor (Supplementary Methods and Supplementary Fig. 21). For this, we used 33 profiles of 126 metabolites, 53 proteins and 75 genes that constitute the core metabolism, in order to predict the concentration of each metabolite. We found that using (measured) protein expression levels leads to better results (PCC $0.65 \pm 0.21$) than by using gene expression levels (PCC $0.47 \pm 0.26$) as shown in Fig. 5d. For predicting concentrations of metabolites in non-core metabolism, we resort to the inference of enzyme concentration from mRNA expression levels due to the paucity of profiles with both metabolome (including metabolites in non-core metabolism) and proteome information (only 6 profiles). Predictions on the metabolome layer are more accurate in non-core than in core metabolism ($P < 10^{-13}$ and $P < 10^{-3}$ for non-core and core metabolome, respectively; Fig. 5d). This is an expected result, given the variability analysis in Ecomics, as concentration variance in core metabolism is high. Variance analysis indicates that the prediction of metabolite concentrations in non-core pathways is robust, with a variance of $0.02 \pm 0.01$, comparable to that of core metabolism ($0.06 \pm 0.01$), which suggests that the non-core metabolic set can be highly predictable even without the presence of protein expression data (Supplementary Fig. 21B). FBA with multi-layered omics data to constraint reactions results in better performance (PCC $0.72 \pm 0.24$) than using plain FBA (PCC $0.65 \pm 0.39$, $P < 10^{-8}$, Fig. 5e, Supplementary Methods) or when one of the layers is absent (PCC $0.67 \pm 0.21$ for proteome only and PCC $0.70 \pm 0.20$ for transcriptome only).

Ultimately, the integrative model had the highest performance overall in predicting growth rate (Fig. 5f and Supplementary Figs 22 and 23; PCC $= 0.65 \pm 0.01$ in leave-one-condition-out cross-validation over 101 conditions with wild-type information present in the training set; 0.76 for novel wild-type cases). Interestingly, although the information added by the inclusion of any given module was not immediately obvious (Fig. 5g), the optimal results were obtained when all modules were present.

**Most informative molecules of expression and growth**. To understand what molecules were significantly involved in the prediction of cellular dynamics, we investigated the features selected (that is, features having non-zero weights) from LASSO constrained regression for each layer of transcriptome, proteome and metabolome. For the transcriptome layer, LASSO selected 733 genes to predict growth rate. Among them, 15 genes were from the cellular machinery of ribosome and RNA polymerase (48 genes total, $P < 0.001$). Gene Set Enrichment Analysis of all 733 genes shows its enrichments in essential gene ontologies including ATP ($P < 5.1 \times 10^{-39}$), cell wall ($P < 7 \times 10^{-31}$) and DNA-replication ($P < 3.7 \times 10^{-5}$). Interestingly, most of the top genes with the highest absolute weight are not directly involved in growth-related processes, but they directly interact with cellular

machinery at the protein level. For example, the third most informative gene, *mreB* has known PPIs with all 48 genes of RNA polymerase and ribosome (Supplementary Methods, for complete list of selected genes, see Supplementary Data 4). Four of the top 10 genes are uncharacterized and can be potential targets for further investigation. In addition, knockout of seven out of the nine most informative genes (Keio collection in LB; *yffO* omitted as not in the library) had moderate to severe negative growth rate effects (Supplementary Table 3). Note however that the prediction performance drops strikingly if we only pick the top ten genes as features, with the PCC decreasing to 0.08 from 0.65, which suggests that the integration of information from multiple processes is needed to predict growth given the high diversity of the possible conditions. For the proteome layer, MOMA selects 14 out of 1001 proteins, enriched in cellular processes directly related to cellular growth including translation ($P < 6.3 \times 10^{-4}$), ribosome ($P < 7.4 \times 10^{-4}$) and tRNA binding ($P < 1.7 \times 10^{-5}$). For the metabolome layer, MOMA identified nine metabolites with non-zero weight, noticeably mainly from amino acids (Table 1). The list of all selected molecules with their absolute weights in three omics layers are in Supplementary Data 4.

**Experimental validation of MOMA and model comparison**. We used the 16 novel genome-wide transcriptional profiles that we performed for Ecomics expansion (see Methods) to evaluate the prediction performance of MOMA. These profiles correspond to conditions that are the least represented in Ecomics, so the performance estimate is expected to be conservative. We used three initialization points for each gene: its average expression over all MG1655, M9/LB profiles (baseline 1, non-specific), its average expression over all BW25113, M9/Glucose profiles (baseline 2, specific) and its average expression over the three BW25113, M9/Glucose replicate profiles that were measured in the same batch with the 16 knockouts (baseline 3, same-batch). In all cases, the prediction was significantly better (27, 41 and 25% from baselines 1, 2 and 3, respectively; Fig. 6a). The PCC of MOMA's prediction in baseline 3 was between 0.58 and 0.85 for the 16 knockouts. Prediction outliers, such as the large discrepancy between the high genome-wide prediction performance of the *kefB* knockout and the expression of the *ydbH* gene in the same sample, can serve as targets for further investigation (Fig. 6b).

We then compared the prediction performance of MOMA to that of the leading *E. coli* prediction methods, ME-Model[26] and EBA[5]. As each method predicts a different set of genes, we compare the expression levels of the gene set predictable by all three methods. Regardless of the initialization used, MOMA significantly outperformed the other two methods (38–280% for ME-Model and 47–311% for EBA) with the performance difference increasing with the information context of the baseline (Fig. 6c).

## Discussion

There are three elements necessary to realize the potential of predictive biology: data, models and methods to improve both. Here we presented Ecomics, a coherent multi-omics database for *E. coli* that can be used for training traditional and advanced machine learning methods. There are a number of lessons learned from this work. First, quality control of high-throughput data is both necessary and challenging, due to the lack of reporting standards and the complexity of any given experimental setting. The extent to which next-generation machine learning techniques[47] can help us solve biological questions depends heavily on the protocols and the standards in place for efficient meta-data reporting and quality control. Even in this case, the creation of a unifying, normalized multi-omics compendium is far from a simple data aggregation, as several integration steps based on biological knowledge are needed. As our analysis showed, current data unevenly explore the experimental space and are difficult to integrate due to the paucity of cross-layer information for the same conditions. Investment in these fronts is required, if we aspire to approach the exploration of the biological experimental space in a structured manner[18] to guide new omics profiling. Future extension of Ecomics includes multiple data types such as ChIP-Seq and connection to other *E. coli* resources such as EcoCyc[32] and RegulonDB[48]. We envision a structured data ecosystem that will provide the data and meta-data necessary for advanced predictive analytics.

The omics integration allows us to reconstruct an ontology of the environmental (stress, medium) and genetic (strains, gene perturbations) features based on their effect on genome-wide molecular expression, which itself is a proxy for several processes and behaviours. Although a network-based GO has been explored before[49], the construction of the Ecomics compendium allowed us to reconstruct ontologies related to environmental settings and

**Table 1 | The top 10 informative features to predict growth rate for each of the three layers.**

| Rank | Gene | Function | Protein | Function | Metabolite | Category |
|---|---|---|---|---|---|---|
| 1 | *wcaF* (0.26) | Predicted acyl transferase | TufB (0.12) | Elongation factor Tu | Cytosine (0.12) | Nucleobase |
| 2 | *yffO* (0.17) | CPZ-55 prophage; predicted protein | Syd (0.04) | SecY-interacting protein | L-glutamine (0.11) | Amino acids |
| 3 | *mreB* (0.09) | Dynamic cytoskeletal protein MreB | AnsA (0.04) | Asparaginase I | L-alanine (0.07) | Amino acids |
| 4 | *yfiP* (0.09) | Conserved protein | ParC (0.04) | Dimer of topoisomerase IV subunit A | L-threonine (0.07) | Amino acids |
| 5 | *tyrP* (0.08) | Tyrosine:H + symporter TyrP | PyrH (0.01) | UMP kinase | Nicotinamide (0.04) | Vitamin |
| 6 | *ycbT* (0.08) | Predicted fimbrial-like adhesin protein | TrxB (0.01) | Thioredoxin reductase | L-valine (0.04) | Amino acids |
| 7 | *gfcC* (0.08) | Conserved protein | GloB (0.01) | Glyoxalase II | Azelaic acid (0.04) | Fatty acids and conjugates |
| 8 | *solA* (0.08) | N-methyltryptophan oxidase | SerS (0.009) | Seryl-tRNA synthetase | D-glycerol-1-phosphate (0.02) | Glycerolipid metabolism |
| 9 | *fecA* (0.08) | Ferric citrate outer membrane porin FecA | RpsL (0.009) | 30S ribosomal subunit protein S12 | L-ornithine (0.008) | Non-proteinogen amino acid |
| 10 | *ynfB* (0.07) | Predicted protein | PotD (0.004) | Putrescine/spermidine ABC transporter | — | — |

LASSO constrained regression was used to select informative features for predicting growth rate. The number in parentheses represents absolute weight and is an indicator of the significance of that molecule.

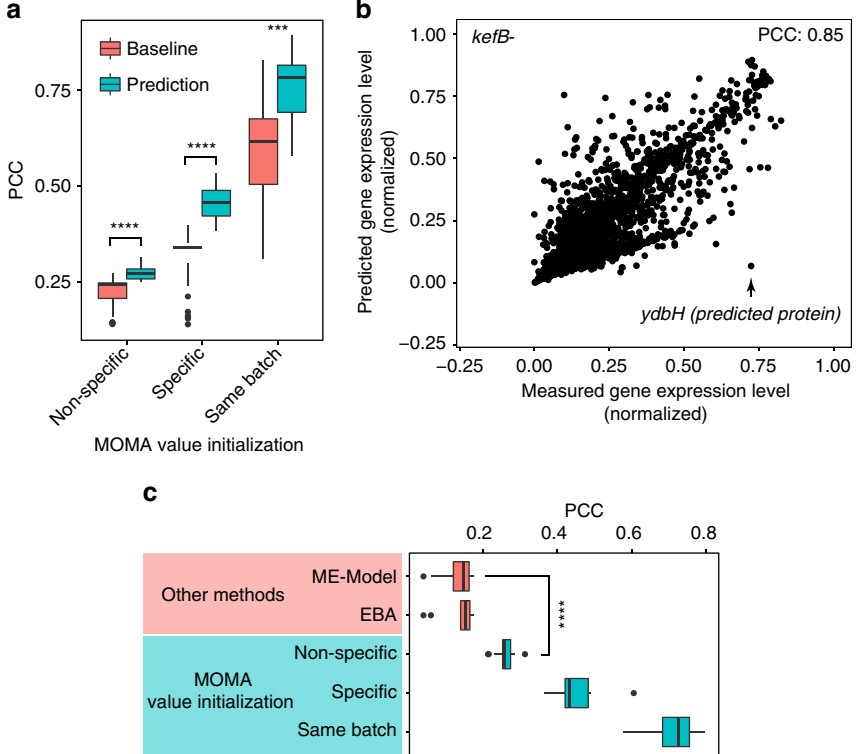

**Figure 6 | Model validation for 16 novel knockouts in transcriptome prediction and performance comparison with other methods.** (**a**) PCC between predicted and measured expressed levels of 4096 genes for 16 knockout conditions. Three expression value sets were used to initialize the recurrent neural network (RNN). Non-specific initialization corresponds to the case where the RNN was initialized with the values of the most frequent condition in Ecomics, MG1655 cells in M9/LB media ($P<10^{-10}$). Specific source initialization corresponds to the mean expression profile of BW25113 cells in M9/Glucose in the compendium, which is the same media and strain that was used in the knockouts ($P<10^{-7}$). Same-batch initialization corresponds to the mean expression genome-wide profiles of BW25113 cells in M9/Glucose that was measured in the same batch with the 16 knockouts ($P<10^{-3}$). (**b**) Scatter plot between predicted and measured expression levels for the kefB knockout condition. (**c**) Method comparison of prediction performance. We compared the prediction performance of transcriptome response to 16 KOs with ME-Model[26] ($P<10^{-6}$ for non-specific, $P<10^{-13}$ for specific, $P<10^{-16}$ for same-batch), and EBA[5] ($P<10^{-7}$ for non-specific, $P<10^{-14}$ for specific, $P<10^{-16}$ for same-batch). The error bar indicates standard error over the mean. ***($0.001<P<0.01$), ****($0.0001<P<0.001$). Wilcoxon rank sum test was used for all statistical tests.

strains. We argue that this is a more informative approach for understanding complex environments than relying on chemical composition or genetic similarity alone. Further work towards this direction can include a multi-trait analysis that will lead to an organism-specific environmental map, which will be useful for understanding bacterial physiology and its evolution.

This work is the first that trains an integrative model in a comprehensive set of all available omics and interaction data across multiple layers; hence, the results are representative of what multi-omics models like MOMA can achieve in a large spectrum of conditions. It is worth noting that in all comparisons, the integration of multiple information always over-performed the individual predictors. This observation also holds in the case of non-core metabolome prediction and genome-wide flux prediction from FBA coupled with the multi-omics data set. The prediction performance, both in cross-validation across the *E. coli* omics universe and in new RNA-Seq and growth experiments that we conducted, is at levels that support the use of this technology for forward predictions in novel environments. At the same time, analyses of the molecules are the most informative for growth prediction, reveal that a different group of biomarkers emerges for each layer. As expected, amino acids are the dominant group for metabolomics, while transporters and growth-related proteins comprise the informative set. In transcriptomics, there are a number of unknown conserved genes, among membrane and transport related genes, which need further investigation.

Given the size and types of data used to build, train and evaluate the predictive model, this study provides a comprehensive view of the possibilities and challenges in data-driven, genome-scale modelling. Integration of additional data sources, such as genome-scale data sets with structural information, have the potential to provide orthogonal information and further increase the prediction performance. Additionally, the targeted experimental refinement is *conditio sine qua non* for the optimal exploration of the experimental space for any particular organism. This can be extended further to include other models, data types and collection of organisms or bacteria consortia, hence providing a further step towards the structured experimentation, exploration and discovery in the era of 'Big data'-driven predictive biology.

## Methods

**Data integration overview.** A total of 4389 genome-wide profiles over 649 conditions were collected from 26 databases and literature curation (291 sources). These omics profiles cover the transcriptome (3579 profiles of 4096 transcripts), proteome (71 profiles of 1001 proteins), metabolome (696 profiles of 356 metabolites) and fluxome (43 profiles of 120 fluxes). We created a detailed meta-data ontology (Supplementary Data 5) that includes 612 features related to genetic information (154 features), experimental settings (52 features), medium chemical composition (120 features) and genetic perturbations (286 features). Raw data of the transcriptome and proteome were processed using different methods depending on the platforms used (that is, RNA-Seq, one-channel array, two-channel array for transcriptome, Mass-Spectrometry for proteome). We only used processed data for the metabolome and fluxome. Once data were processed to quantify concentrations of molecules, we applied the same normalization principles

based on semi-supervised approaches to three layers of transcriptome, proteome and metabolome as follows: Noise content was removed from the observed expression level by a Gaussian mixture model. Platform-biases were corrected by applying platform-specific $Z$-score normalization. Absolute-level quantification was performed by loess regression between known absolute-level expression levels and relative expression levels. For processed fluxome data from multiple sources, we normalized reaction rates by the glucose uptake rate. Additional growth dynamics (plate-readers; 1996 profiles) were experimentally measured and annotated through automated procedures. An overview of the normalization procedure is depicted in Fig. 2.

**Transcriptional layer.** *Raw-data pre-processing.* Each of the three major platforms (one channel, dual channel and RNA-Seq) was processed by specialized methods or in a separate pipeline. For one-channel array, image data were first read into R using the affy package. Then RMA[50] is applied for a set of replicates for background correction, normalization, probe-set summarization. For dual-channel array, image data were read into R using the limma package. Background correction was performed using normexp (with an offset of 5) as it is shown to outperform other methods[51]. Red and green channels were separated and quantile-normalized for each set of replicates.

The publicly available RNA-Seq data were downloaded from Sequence Read Archive, first converted to fastq using fastq-dump and then processed to follow the process (that is, Trimmomatic, TopHat/bowtie2, htseq-count) below. The low qualities of raw reads were trimmed using Trimmomatic (v0.30)[52] with default settings. Trimmed reads were aligned on the most recent reference genome of *E. coli* K-12 MG1655 (GenBank: U00096.3) by using TopHat (v2.0.10) coupled with bowtie (v1.0.0)[53]. The resulting SAM file is then processed to have gene-level read counts using htseq-count[54]. From the 2346 raw profiles, we finally have 2329 profiles after discarding unreadable raw files (for example, custom-designed arrays without array design specification).

*Noise removal.* Noise in gene expression was estimated from three distinct sources, phantom genes[32], negative-control probes designed by array manufacturers and non-K12 strain genes. Means and variances of both the signal and the noise were estimated using the Expectation Maximization (EM) algorithm implemented in R mclust, with the intensity below mean intensity of the three noise sources considered as noise during initialization. Gene expression was then updated throughout the Ecomics compendium.

*Platform-bias correction.* To correct systematic biases due to different technology platforms[55], we applied platform-specific z-score transformation[56]. Most platforms are designed to measure relative expression levels, complicating the direct integration of profiles across different studies. For this reason, we first performed quantile-normalization (using R package preprocessCore) for each platform and then we transformed the expressed genes on the same scale for each profile, by using loess fit between expression levels of before and after quantile-normalization. Then finally, we applied a z-score transformation for each platform data set (Supplementary Figs 24A,B).

*Absolute-level quantification.* Due to fluctuations in the total RNA/cell[16], it is assumed that uniform expression distributions across different conditions can lead to inaccurate downstream analysis[21]. To avoid this issue, we converted the relative expression measurements to absolute RNA copies per cell, by applying loess regression between the measured expression level and the absolute expression level for each profile. In cases where we have relative (that is, Ecomics profiles after platform-bias correction) and absolute expression levels[44] for some genes (the 'shared genes'), we trained a loess regression model, which was applied to the rest of the genes, and the process was repeated for all profiles[21]. Focusing only on housekeeping genes as a reference produced inferior results compared to this method. We compared normalized expression levels between genes with short half-life ($0.573 \pm 0.004$) and genes with long half-life ($0.572 \pm 0.003$). The results (Supplementary Fig. 24C) show that the mean difference is not statistically significant ($P = 0.41$).

*Dealing with missing expression levels.* In Ecomics, genes that had more than 70% of their expression values missing were removed from the compendium. Similarly, we removed profiles where more than 70% of gene values were missing. Through this process, 502 genes and 2 profiles were excluded from Ecomics. For the rest of the profiles/genes, we imputed their values by applying for each gene a method that is based on the $k$-nearest neighbours ($k$ is 3 here) algorithm (R package impute).

The final Ecomics transcriptional layer that resulted after following this methodology had 4096 genes over 3579 profiles (February 2016).

**Protein layer.** A total of 71 proteome profiles were downloaded from ProteomeXchange and PRIDE databases[57,58], as well as other published sources (Supplementary Data 6). We also experimentally profiled six samples in the transcriptome and proteome layers since the overlap between these layers is low (Supplementary Methods). Similar to the transcriptomics layer, we filtered out profiles for non-K12 strains and evolved strains. Mass spectrometry raw files were first processed by MaxQuant[59] with default settings, with the *E. coli* K-12 protein sequences downloaded from UniProt[60] (Tax ID: 83333) to map peptides to protein sequences. Absolute intensities (iBAQ) were then produced[61]. For processed profiles that contain relative protein concentrations, we transformed them to

absolute protein copy number per cell, by matching relative concentrations with APEX absolute protein numbers from ref. 45 and then performed loess regression. The missing values were imputed, as described above. Special attention was given to the processing of the profiles reported in ref. 62, a high-quality data set of 59 proteins in 34 profiles, which was imputed separately, filtering out 4 of these proteins across the profiles. In all other cases, this step filtered out 539 proteins, which led to a final size for the proteomics layer of 71 profiles (69 profiles with exponential phase) with 1001 proteins.

**Metabolic layer.** A total of 696 processed profiles were obtained from literature curation (Supplementary Data 6). For quantifying absolute metabolite numbers per cell for each metabolite, profiles having molar concentration were normalized as in ref. 63, yielding $\widetilde{m}_j^i = 10^6 \cdot m_j^i / \sum_{j=1}^n m_j^i$ where $\widetilde{m}_j^i$ is the number of molecules for metabolite $j$ in profile $i$, $m_j^i$ is concentration, $10^6$ is the normalizing constant, assuming that concentration of 1 nM in an *E. coli* cell corresponds to 1 molecule in the cell[64]. For metabolites with both measured concentrations and absolute molecule numbers, we fitted a loess curve and then used it to convert all metabolites in the corresponding profile[63]. The missing values were imputed, as described above. To ensure consistent naming of the metabolites, we used PubChem[65], ChemSpider[66], KEGG[67], EcoCyc[32], HMDB[68] to resolve naming issues. As it was the case in the proteome layer, we processed separately[62], which resulted in 138 metabolites excluded, finally producing 33 profiles of 126 metabolites. In all other cases, 71 metabolites were excluded, resulting in a metabolic layer of 696 profiles for 356 metabolites.

**Fluxomics layer.** We collected publicly available fluxome data from the literature. A total of 43 profiles of 120 fluxes were compiled (Supplementary Data 6). As all profiles were in experimental conditions with glucose as the main carbon source, all reaction rates were normalized based on glucose uptake rate for each profile ($f_j^i = 100 \cdot f_j^i / f_{glu}^i$) where $f_j^i$ is the absolute flux of reaction $j$ for profile $i$ and $f_{glu}^i$ denotes the measured glucose uptake rate for profile $i$. As different synonyms of a reaction were present across different studies, reactions were cross-referenced in KEGG[67] and BiGG[69] databases.

**Phenomics layer.** *Growth experiments.* We were able to collect growth rate meta-data for 767 profiles (17.6% of all profiles, growth rates ranging from $0.01\,h^{-1}$ to $2.14\,h^{-1}$). However, we noticed that growth rates were inconsistent even for the same conditions due to differences in instruments, measurement protocols, among others. For this reason, we re-created the experimental conditions for 50.3% (48% for exponential phase, 58% for stationary) of all profiles in the compendium and systematically measured growth rates with plate-readers, after they were rectified for offset minimization. For the phenomics layer, the growth curves that were experimentally measured (Biotek Synergy HT, 96-well plates in triplicates) used an automated script and consistent thresholds to identify the growth characteristics. An automated MATLAB script was used to calculate the lag phase, maximum growth rate and maximum cell density. The lag phase is defined as the time until 5% of the maximum growth rate is reached. Growth rates were determined by calculating the differential between 10 and 90% of cell density using a virtual sliding window with the width of 1 h and sliding every 15 min. This led to experimental growth dynamics measurements for another 1992 profiles (out of 3579; 55.6% of all profiles) with transcriptional information, corresponding to 179 conditions (out of 596; 30% of all conditions). Similarly, we measured the growth dynamics for another 57 (33 conditions) and 577 profiles (49 conditions) that covers 80.2, 82.9% of profiles with proteomics, metabolomics and fluxomics information, respectively. All measured growth data are available in Supplementary Data 7.

*Inference of un-annotated growth phase.* A classifier was trained to identify exponential and stationary phase, based on genome-wide gene expression (no profiles were in lag phase, as cells do not grow). SVM classification through an RBF kernel was found to have the highest performance. Features were selected by standard wrapper techniques with features (genes) being first ranked by their mutual information to the growth phase and then selected sequentially by cross-validation performance. To assess the prediction performance of our method, we used all profiles with high-quality information on phase, yielding a data set of 458 profiles with an equal fraction of stationary phase and exponential phase (out of 3579 profiles total, only 229 were in the stationary phase). We then performed a tenfold cross-validation, producing a Receiver Operating Characteristic and Precision-Recall curve (Supplementary Fig. 25). By using this technique, we labelled 1493 profiles (101 conditions) that were missing growth phase metadata.

**Model.** *Input and output.* We built a genome-scale MOMA platform that aims to predict an output vector $y$ containing the expression levels of mRNAs (4096), proteins (1001), metabolites (356), metabolic fluxes (2382) and growth dynamics, for a given experimental condition. The experimental condition is represented as an input vector $x$ with 612 features that contain information about the genetic (strain, genetic perturbation) and environmental (medium, stress) background (Supplementary Methods). For initializing the model during training, the averaged expression profile of MG1655, in M9/LB was used.

**Modularized architecture.** MOMA is designed to be modular, in order to allow for a better representation of biological organization, an easier manipulation of individual modules and to avoid dimensionality issues. The final model is modularized into multiple layers of input, transcriptome, proteome, metabolome, fluxome and phenome. For each module, we evaluated various statistical techniques in their ability to capture biological structure and make accurate predictions.

**Model training data.** Our analysis focused on profiles for samples in the exponential phase, as the amount of data is sufficient for a rigorous assessment of the model. The data resources of Ecomics compendium and all the network resources (Supplementary Table 4 for Sigma factor network) used for model training are summarized in Fig. 1.

**Transcriptome layer.** For the transcriptional layer, an RNN[42] with sigmoid activation functions that takes the 612 features as inputs and predicts the expression of 4096 genes from 612 features was trained using $l_1$ regularization and stochastic gradient descent (Supplementary Methods). In RNN, each connection $c_{i,j}$ between a node $i$ and a node $j$ has a weight $w_{i,j}$ associated with it (Supplementary Fig. 13). We represent the weights for connections between input nodes $x$ (612 nodes) and output nodes $y$ (4,096 nodes) as $w_x$ and the weights for connection among output nodes $y$ as $w_y$, so that the set $W = \{w_x, w_y\}$ includes all weights of the RNN. Given input $x$ and weights $W$, the output vector $y$ is iteratively computed by:

$$y^{(i)} = h\left(w_x x + w_y y^{(i-1)}\right)$$

for $1 \leq i \leq n$, where $y^{(i)}$ is the state of output vector $y$ at iteration time $i$ and $n$ is the memory depth. During the training phase, $W$ is adjusted by minimizing the residual sum of squares between observed $y$ and predicted $y$ for all training data based on stochastic gradient descent. All the hyper-parameters are empirically optimized. The optimal memory depth was 2 and the cycles having length less than 3 account for 75% of all cycles in the TRN in *E. coli* (Supplementary Fig. 26).

**Proteome layer.** The protein expression values of 1001 proteins were predicted using LASSO constraint regression through a consensus of the transcriptional regulatory, PPI, co-expression network and other pathway information (for more information, Supplementary Methods). More specifically, for Transcriptional Regulatory Network (TRN), it was built based on the RegulonDB database. The protein level of a target gene in a novel condition is predicted by LASSO regression of the expression levels of genes that are connected through a regulatory link to the target gene. For Protein-Protein Interaction (PPI) Network, it was constructed from the five distinctive sources that were described in Supplementary Methods. Similarly, the protein level of a target gene in a novel condition is predicted by LASSO regression of the expression levels of genes whose respective proteins are connected through a Protein-Protein interaction to the target protein. For Co-expression Protein Network (CPN), it (70,710 interactions of 3,163 proteins) was built from the core proteome dataset (Supplementary Methods), which represents 20 expression profiles of 1,001 proteins, which are not used for proteome prediction. For two proteins to be considered co-expressed, their pairwise correlation should be larger than 0.7. Any given protein level in a novel condition is predicted by LASSO regression of the protein expression levels of co-expressed proteins with respect to the target protein. For pathway clustering, we cluster genes that are implicated in the same pathways, as represented in the KEGG database. The protein level of a target gene in a novel condition is predicted by LASSO regression of the expression levels of genes that are implicated in the same pathways as the target gene.

**Metabolome layer.** The concentrations of 356 metabolites are predicted from 4096 genes in the transcriptome layer (for non-core metabolism) and 1001 proteins in the proteome layer (for core metabolism) using regression with $l_1$ regularization. For metabolites having known enzyme-substrate relations, we predict its concentrations from the expression levels of the related enzymes. For those with no such information, we fit from all the genes by using LASSO, which allows variable selection (Supplementary Methods).

**Fluxome layer.** FBA was used to predict 2382 fluxes, while protein/transcript and extra-cellular information from three layers of transcriptome, proteome and input was used to inform bounds. More specifically, lower bounds of reactions changed by:

$$\begin{cases} l_i = 0 \text{ if } \text{mean}(g_i) \leq t \\ l_i = -1000 \text{ otherwise} \end{cases}$$

where $l_i$ is expression levels of enzymes in reaction $i$. $t$ is empirically determined by finding optimal parameter with maximum predictive performance within the predefined range (for more information, Supplementary Methods). Bounds for all exchange reactions are listed in Supplementary Data 8.

**Phenome layer.** Final growth rate prediction was calculated by the weighted sum of predicted growth rates from each individual layer, with the weight proportional to the performance of the respective layer during cross-validation and each layer predicts growth rate from concentration of molecules in the layer as well as extra-cellular information based on LASSO constrained regression (for more information, Supplementary Methods and Supplementary Fig. 27).

**Model performance and experimental validation of predictions.** We evaluated the performance of the model by performing a leave-one-condition-out cross-validation for each layer (more information in Supplementary Methods and

Supplementary Fig. 28 for transcriptome layer). Additionally, we performed transcriptional profiling for 16 single-gene knockout strains selected based on GO coverage optimization (Supplementary Methods). During model training we evaluated three distinct RNN initialization methods: using the mean expression profile in Ecomics of MG1655 with M9/LB media (the condition with the most profiles in Ecomics), the mean expression profile in Ecomics of BW25113 with M9/Glucose (the condition that follows the experimental setup), and the mean expression profile in the new experimental batch of BW25113 with M9/Glucose. To evaluate the effect of the gene knockout on expression, we set the expression of that gene to zero in the RNN, and set the other input parameters according to the experimental conditions (genetic features of the BW25113 strain and M9/Glucose as the medium).

**Comparison across genome-wide prediction models.** We compared the genome-wide performance of the MOMA model with two other recently published models, the ME-Model[26,70] and EBA[5]. For simulating the gene knockout conditions (16 KO genes) in the ME-Model, we followed the instructions provided in the supplementary material of refs 26,70. For executing the EBA algorithm to predict expression in a gene knockout condition[5], we set the following parameters: flag_topo = 1 (which uses RegulonDB and inferred interactions), flag_mod = 1 (M9) and genetic_pert is set to have -1 for the corresponding KO gene. The ME-Model and EBA methods can predict the expression of 730 (approximately) and 4189 genes, respectively, so the comparison among the three methods (ME-Model, EBA and MOMA) is on the union of the three sets (730 genes, approximately). For each method, the distance between the predicted expression and the mean replicate expression is calculated and the PCC is used for the comparison across all genes.

**Data availability.** The Ecomics compendium and the predictive model is available at http://prokaryomics.com as an online resource. The RNA-Seq data produced from the lab is available at the National Center for Biotechnology Information Gene Expression Omnibus (NCBI-GEO) under the accession GSE73673. The rest of the data that support the findings of this study are available from the corresponding author.

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

## Acknowledgements

We would like to thank Saeed Tavazoie, Bernhard Palsson, Adam Feist, Hani Goodarzi and Justin Siegel for their suggestions and informative discussions. We would like to thank Courtney Roper for proof reading the manuscript. Computational resources and HPC support was provided by the NCSA Blue Waters supercomputing team. This work was supported by ARO award W911NF1210231 and NSF awards 1254205 and 1516695 to IT.

## Author contributions

I.T. conceived and supervised all aspects of the project. M.K. constructed the Ecomics compendium and the integrative model. V.Z. performed the omics experimental profiling. N.R. performed the experimental growth characterization. I.T. and M.K. analysed the data and wrote the paper. All co-authors have read and approved the manuscript.

## Additional information

**Competing financial interests:** The authors declare no competing financial interests.

