## [Peer Review File · Nature Communications]

Reviewer #1 (Remarks to the Author)

The authors describe a significant amount of work to build a multi-omics database for bacteria, then describe a pipeline for cross-data-type and within-data-type normalization (a key step), and lastly describe a method for combining this data into a coherent model for *E. coli*. Overall I think the work is important and well done. I do however find the several problems with how the paper was written that need to be addressed.

The work is focused on *E. coli*, are you doing this for other organisms? If not then *E. coli* should be in the title.

Can you better discuss and highlight the differences between the transcriptomics and proteomics data normalization steps. For transcription I thought the description of what you did was more complete than the description of your pipeline for the other datatypes.

What other methods could you explicitly compare to? Is a range of data prediction of 0.54 to 0.87 good?

In addition to error on model output I'd like some more discussion of model stability / error on model components.

MOMA was not described well enough. You need to add more explicit detail and also restructure both the methods and main results sections on MOMA. I found the methods section quite confusing.

Rely less on the names of your programs in the early intro and abstract where they are, essentially, undefined terms.

I was not 100% clear on how much of the data-prediction in a single layer relied on other layers. For example, how would a MOMA with the layers split into separate prediction tasks (for transcription, protein, and metabolite) perform. This could be a critical comparison.

Your absolute transcript level normalization (partially supervised) seemed quite interesting and novel, can you emphasize this a bit more?

Some of your discussion of GO is at odds with the prediction of data themes and needs better connection sections to prevent readers from getting a bit confused there. Go back over the sections that discuss GO and KEGG ontologies.

Great work!

Reviewer #2 (Remarks to the Author)

This manuscript is a brave attempt to integrate proteomics, transcriptomics and metabolomics done by different groups in *E. coli* in a single source that could be used with the appropriate tools

for predicting the response of the bacteria to perturbations. The difference with previous attempts is that the authors have curated the datasets and normalize the data so as to be able to compare results obtained from different groups. Using this clean dataset and their MOMA analysis they predict transcriptional changes for 16 experiments they have performed, as well as protein and metabolite levels, and growth rates. Compared with existing methodology they seem to do better.

I have some main concerns regarding the manuscript. The first one is about the normalization of transcriptomics. As the authors acknowledge the assumption that total RNA does not change per experiment is not correct and there could be conditions where RNA synthesis is reduced and degradation predominates. In the method section they claim they have solve this problem by applying loess regression between the measured expression level and the absolute expression level for each profile. I think that although this decreases the problem it does not completely eliminate it. In this respect why the authors dont use only fold changes with respect to the reference in the experiment?. From what they say using housekeeping genes produces inferior results. For example they could check if RNAs with very short half lifes or RNAs with long half lifes behave differently in their normalization, this could suggest that there is a problem with the normalization. Regarding proteomics they acknowledge that RNA and protein levels correlate poorly as observed in other bacteria. However, when they combine this with protein complexes, transcriptional networks, protein co-expression and pathway information. I find this section confusing, do they mean they improve the correlation between protein and mRNA or just predict more proteins being expressed for a particular condition? The first case will be remarkable, the other in a way is logical. This is a general remark for all the manuscript. How much of what they predict is due to the datasets incorporated? A good validation and possible within the time of a review is to perform KOs or overexpression of some of the top 10 genes with unknown function and being most informative for growth rate, ie mreB and the 4 uncharacterized ones from the top 10. This will provide a strong validation of the predictive power of their integration and approximation.

Thus i think this is an interesting piece of work, it shows its validity when making predictions for transcriptomics, metabolimics and proteomics but my doubt aside from normalization procedures is how much novel prediction capacity it has. For this i think mutating or overexpressing some of the most informative genes fro growth with unknown function could be an excellent way of validating the methodology.

Reviewers' comments:

Reviewer #1 (Remarks to the Author):

The authors describe a significant amount of work to build a multi-omics database for bacteria, then describe a pipeline for cross-data-type and within-data-type normalization (a key step), and lastly describe a method for combining this data into a coherent model for *E. coli*. Overall I think the work is important and well done. I do however find the several problems with how the paper was written that need to be addressed.

1. The work is focused on *E. coli*, are you doing this for other organisms? If not, then *E. coli* should be in the title.

We have changed the title of the manuscript to “Multi-omics integration accurately predicts cellular state in unexplored conditions for Escherichia coli”.

2. Can you better discuss and highlight the differences between the transcriptomics and proteomics data normalization steps. For transcription I thought the description of what you did was more complete than the description of your pipeline for the other datatypes.

The main difference is the pre-processing to convert raw data to molecular concentration. For raw transcriptome data, we applied different methods to quantify expression levels, depending on the platforms used (i.e. RNA-Seq, one-channel array, two-channel array). For raw proteome data, we applied the MaxQuant tool that processes the raw Mass-Spectrometry data to concentrations of proteins. For other omics data (metabolome and fluxome), we only used processed data.

Once the raw data are processed to depict molecular concentrations, we applied the same normalization principles described extensively in “Transcriptional Layer in Methods” to data in all layers. We have revised the methods section (lines 411-417) to clarify this point.

3. What other methods could you explicitly compare to?

We compared our method MOMA with ME-Model (O'Brien et al. 2013) and EBA (Carrera et al. 2015) for prediction of transcriptome response to novel 16 knockouts. The results of the performance comparison are detailed in the Results (Section: “Experimental validation of MOMA and model comparison”), especially lines 336 to 341. To add more clarity we also revised the introduction to specify that the two methods we explicitly compared (line 105-107).

4. Is a range of data prediction of 0.54 to 0.87 good?

*There is always room for improvement (given more data, incorporation of other data types, etc.), but we are very happy with this result. The prediction performance is encouraging, given that (a) we perform this for all publicly available data for *E. coli*, not just a subset as is usually the norm, (b) it is significantly higher than all baselines: in every single layer,*

the reported numbers are significantly higher than the baseline predictions (based on random, mean and WT profiles) for expression and molecular concentration. The random baseline represents prediction from mean expression level of randomly selected profiles in the training set; the mean baseline is the prediction from mean expression level of all profiles in the training set; the wild-type baseline is the prediction by mean expression levels of wild-type profiles in the training set. In all cases, leave-one-condition-out (LOCO) cross validation was used.

In the transcriptome layer, PCC of 0.54 ± 0.15 was achieved for prediction of genome-wide expression levels and this was statistically significant than three baselines of random (PCC of 0.25 ± 0.17 , $P < 10^{-15}$), mean (PCC of 0.26 ± 0.14 , $P < 10^{-15}$), and wildtype (PCC of 0.36 ± 0.22 , $P < 10^{-15}$).

For prediction of the proteome layer, the consensus approach we used in MOMA performs PCC of 0.55 ± 0.26 , which was statistically significant than two baselines of random (PCC of 0.24 ± 0.34 , $P < 10^{-2}$) and mean (PCC of 0.16 ± 0.10 , $P < 10^{-4}$).

For fluxome prediction, MOMA performs (PCC of 0.72 ± 0.24) statistically significantly higher than mean baseline PCC of 0.50 ± 0.11 ($P < 10^{-8}$).

In the metabolome layer, PCC of 0.65 ± 0.21 and 0.87 ± 0.15 was achieved for predicting concentrations of metabolites in core and non-core metabolism, respectively, which were all statistically significant than random baseline (PCC of 0.36 ± 0.13 , $P < 10^{-6}$ for core and PCC of 0.77 ± 0.17 , $P < 10^{-13}$ for non-core) as well as mean baseline (PCC of 0.47 ± 0.19 , $P < 10^{-3}$ for core and PCC of 0.70 ± 0.13 , $P < 10^{-15}$ for non-core).

*We have revised the abstract as well as the captions for **Fig. 5** to clarify that the reported performances are statistically significant with respect to the baselines.*

5. In addition to error on model output I'd like some more discussion of model stability / error on model components.

*The error of each layer is depicted and reported in **Fig. 5**. We have now added an analysis that shows that this error is pretty stable and only slightly changes when predicting genes in different biological pathways (**Fig. S20**).*

*To investigate how the size of the data impact the model performance, we randomly selected 10%, 25%, 50%, 75%, 90% of the data (w.r.t. the whole Ecomics), run cross-validation and evaluated the model performance in these reduced datasets. We repeated this procedure 10 times (sampling with replacement). As shown in **Table S3**, the overall prediction performance increases from a PCC of 0.47 ± 0.15 to 0.64 ± 0.15 when data size increased from 10% use of total data to use of all data (15% increase in mean PCC). Furthermore, for the same (i.e. overlapping) conditions across datasets with different sizes, prediction performance is increasing from PCC of 0.56 ± 0.12 to PCC of 0.69 ± 0.06 (23% increase in mean PCC). We have now added this results in the manuscript (lines from 228 to 230).*

6. MOMA was not described well enough. You need to add more explicit detail and also restructure both the methods and main results sections on MOMA. I found the methods section quite confusing.

We thank to the reviewer for this comment. We have revised the manuscript (in the main results sections on MOMA and the methods) to avoid confusions from readers.

7. Rely less on the names of your programs in the early intro and abstract where they are, essentially, undefined terms.

We have revised the manuscript (in the abstract and the early intro) and main figures (Fig. 1 and Fig. 3) to remove the name of our programs early on (we still refer to Ecomics in the abstract once).

8. I was not 100% clear on how much of the data-prediction in a single layer relied on other layers. For example, how would a MOMA with the layers split into separate prediction tasks (for transcription, protein, and metabolite) perform. This could be a critical comparison.

*To clarify this point, consider the prediction at each layer separately by focusing on **Figure 5A**. For the transcriptional layer, the model's predictions rely solely on the input layer (strain, medium, stress, genetic perturbation). For the proteome layer, predictions rely on the predictions of the transcriptome layer and the 4 interaction networks (TRN, PPI, CPN, Pathway). There are only 18 profiles that include both transcriptome and proteome data (as shown in **Fig. 4A**) and only 5 different conditions, which makes the uncoupled assessment of the proteome module very difficult and cross-validation of the proteome-only model would not produce statistically reliable results since there is only one out of the five conditions that had common features in the training (i.e. strain, medium, stress, genetic perturbation). The metabolome layer depends on the predictions of both proteome and transcriptome and again we cannot perform a cross-validation as we only have 6 profiles that have data for both (intersection at **Fig. 4A**), each with different types of metabolites. For the fluxome, we use the different layers to constrain the FBA analysis, so to decouple it from these layers, we removed them as constraints. Our results show that prediction performance of reaction rates is better when both transcriptome layer and proteome layer are used as constraints (PCC 0.72 ± 0.24) than none (PCC 0.65 ± 0.39) or each separately (PCC 0.67 ± 0.21 for proteome only and PCC 0.70 ± 0.20 for transcriptome only). We have added this analysis in the "Metabolome layer and fluxome layer" section, lines 283-287 and **Fig. 5E**.*

9. Your absolute transcript level normalization (partially supervised) seemed quite interesting and novel, can you emphasize this a bit more?

We have now revised the manuscript with more detailed discussions of the normalization method (abstract; line 3, introduction lines; 91-96). We have also added the new results

that compares prediction performance of the models using either of two different normalization approaches (absolute-scale semi-supervised normalized data versus fold-change data) to support the efficacy of the proposed normalization approach (Fig. S21, manuscript lines from 210 to 214).

10. Some of your discussion of GO is at odds with the prediction of data themes and needs better connection sections to prevent readers from getting a bit confused there. Go back over the sections that discuss GO and KEGG ontologies.

We have revised the result section “Targeted experimentation in Ecomics” (lines 118-133)

Great work!

We would like to thank the reviewer for his/her excellent comments and recommendations!

Reviewer #2 (Remarks to the Author):

This manuscript is a brave attempt to integrate proteomics, transcriptomics and metabolomics done by different groups in *E. coli* in a single source that could be used with the appropriate tools for predicting the response of the bacteria to perturbations. The difference with previous attempts is that the authors have curated the datasets and normalize the data so as to be able to compare results obtained from different groups. Using this clean dataset and their MOMA analysis they predict transcriptional changes for 16 experiments they have performed, as well as protein and metabolite levels, and growth rates. Compared with existing methodology they seem to do better.

I have some main concerns regarding the manuscript.

1. The first one is about the normalization of transcriptomics. As the authors acknowledge the assumption that total RNA does not change per experiment is not correct and there could be conditions where RNA synthesis is reduced and degradation predominates. In the method section they claim they have solved this problem by applying loess regression between the measured expression level and the absolute expression level for each profile. I think that although this decreases the problem it does not completely eliminate it. *In this respect why the authors don't use only fold changes with respect to the reference in the experiment?*

We thank to the reviewer for this comment. We have now performed comparative analysis of prediction performance between two different training datasets; One is based on fold-change and another one is based on absolute normalization. Fold-change data was collected from the compendium COLOMBOS v3 whereas absolute-normalized data was from the compendium Ecomics (this work). We extracted the overlapping conditions between two sources. Absolute-normalized data utilizes the expression levels of reference and perturbed conditions separately unlike the fold-change data that only represents relative differences in expression level. Then the same cross-validation experiments of Recurrent Neural Network (RNN) we designed (Supplementary Text; Section 3.3) were performed based on each of two different sources separately. The results (Fig. S21) show that the prediction performance of absolute-scale data ($PCC\ 0.58\pm 0.16$) outperforms that

of fold-change data ($PCC\ 0.19\pm 0.26$). The baseline performance of fold-change data (wildtype baseline - $PCC\ 0.005\pm 0.04$) was much lower than that of absolute-scale data (wildtype baseline - $PCC\ 0.28\pm 0.07$), mostly ranging in zero (note: \log_2 of fold-change was used). Considering this, we compared the difference in mean PCC between the model prediction and the wild-type baseline and it was much larger in the absolute-scale data ($\Delta PCC\ 0.30$) than in the fold-change data ($\Delta PCC\ 0.185$). We have now added the results in the manuscript in *lines from 210 to 214*.

2. From what they say using housekeeping genes produces inferior results. For example, they could check if RNAs with very short half-life or RNAs with long half-life behave differently in their normalization, this could suggest that there is a problem with the normalization.

We now compared normalized expression levels between genes with short half-life (0.573 ± 0.004) and genes with long half-life (0.572 ± 0.003). The results (**Fig S1C**) show that the mean difference is statistically insignificant ($P = 0.41$). We have added the results in the manuscript (*lines 467-470*).

3. Regarding proteomics, they acknowledge that RNA and protein levels correlate poorly as observed in other bacteria. However, when they combine this with protein complexes, transcriptional networks, protein co-expression and pathway information. I find this section confusing, do they mean they improve the correlation between protein and mRNA or just predict more proteins being expressed for a particular condition? The first case will be remarkable, the other in a way is logical.

It is the first and we have revised the manuscript (lines 258-263) for clarity.

4. This is a general remark for all the manuscript. How much of what they predict is due to the datasets incorporated?

*A very good point that was raised by both reviewers. We have performed new analysis that shows changes in prediction performance in the transcriptome layer with respect to size of training dataset. To create a test dataset, we randomly select 10 times the profiles at the amount of a specific percentage (ranging from 10%, 25%, 50%, 75%, 90%) of the total profiles in the original Ecomics compendium. We run the cross-validation experiments for each dataset. As shown in **Table S3**, the overall prediction performance is modestly increasing from PCC of 0.47 ± 0.15 to 0.54 ± 0.15 when data size increased from 10% use of total data to use of all data (15% increase in mean PCC). Furthermore, for the same conditions across datasets with different sizes, performance of predicting such conditions is increasing from PCC of 0.56 ± 0.12 to PCC of 0.69 ± 0.06 when % of filter-in of dataset increased from 10% to 100% (23% increase in mean PCC). This suggests data incorporation indeed adds predictive ability in the model. We have now added this results in the manuscript (*lines from 228-230*).*

5. A good validation and possible within the time of a review is to perform KOs or overexpression of some of the top 10 genes with unknown function and being most informative for growth rate, i.e. mreB and the 4 uncharacterized ones from the top 10. This

will provide a strong validation of the predictive power of their integration and approximation. Thus I think this is an interesting piece of work, it shows its validity when making predictions for transcriptomics, metabolomics and proteomics but my doubt aside from normalization procedures is how much novel prediction capacity it has. For this, I think mutating or overexpressing some of the most informative genes from growth with unknown function could be an excellent way of validating the methodology.

*We would like to point out that being informative for growth doesn't mean necessarily that knocking out that gene will confer a larger than normal growth effect, although we expect some correlation to exist. To investigate this point, we used the Keio collection and measured the growth rate for strains with a single knockout for the 10 most informative genes (with 6 being predicted or conserved genes) for growth that we identified (single knockouts over the BW25133 WT) and compared it with the growth rate of BW25133 wildtype. The yffO gene (which is a predicted gene) was omitted in the experiments as it is not in the Keio collection. To avoid effects due to medium adaptation and stress, we used LB as the medium. As provided in **Table S4**, the knockout of 7 out of the 9 most informative genes had moderate or severe negative effects in growth rate. For the other two genes, gfcC (conserved gene) and fecA, changes in growth rate were more subtle (0.35%) compared to that of wildtype, implicating two cases might either be false positives or have combinatorial effects beyond what can be detected in a single knockout experiment. We have revised the manuscript to include these results (lines 309-312).*

This review comment, led us to investigate another interesting point regarding the most informative genes. The growth rate predictor has 733 genes with non-zero weight. We investigated if by just keeping the 10 top genes we can actually be predictive. In fact this is not the case, with the performance dropping from a PCC of 0.65 (733 gene features) to 0.08 (the top 10 gene features only), arguing that growth prediction in such a diverse and large condition space (Ecomics), we need to combine information from multiple genes to predict accurately across the board (lines 312-315).

*Finally, we note that MOMA depends on its training set to predict new conditions, so the attributes (strain, medium, stress, genetic perturbation) of such condition should exist, independently or as a combination, to at least one entry of Ecomics, i.e. the training set here (lines 189-197). Since none of the 10 top genes are present in the vector of 286 genetic perturbations in the training set, MOMA should not be used to make accurate predictions (we did validate experimentally forward predictions that adhere to these criteria, as we described in lines 321-342). One additional analysis that we did to elucidate what we are expected to predict with the mean PCC or higher, was to rank the GO categories and KEGG pathways that had higher PCC than the mean. These categories range from 56%-97.5% (lines 223-228) and can be found in **Supplementary Data S8**.*

We would like to thank the second reviewer for his time and valuable comments.

Reviewer #1 (Remarks to the Author)

I had many detailed comments in my initial review that were addressed by the authors in full. Many of my initial comments were requesting additional detail in methods description and requests that the writing be revised. These have been all met with additional computational experiments presented by the authors and substantial edits.

I am in favour of publishing this with minor revisions.

Reviewer #2 (Remarks to the Author)

I have read the manuscript and i found the answers to my questions acceptable. I have only one comment left. The authors state in the proteomics analysis:

"As shown in Fig. 5C, the integration of four methods has the highest protein coverage 242 and can predict all 1001 proteins, while three of the four individual methods can predict a substantial lower number of proteins (250 for TRN, 547 for KEGG, 1,000 for PPI, 847 for 244 CPN). The prediction performance of the integrated method (0.55 ± 0.26) outperforms all 245 individual methods (0.41 ± 0.23 for TRN; 0.47 ± 0.23 for KEGG; 0.48 ± 0.26 for PPI; 0.52 ± 0.24 for CPN)". I will argue that the difference between 1001 and 1000 (PPI) is not substantially different. Similarlry the outperformance is not large (ie 0.55 ± 0.26 vs 0.52 ± 0.24 for CPN). I think the authors should be more modest in their adjectives in this section.

Reviewer #1 (Remarks to the Author):

I had many detailed comments in my initial review that were addressed by the authors in full. Many of my initial comments were requesting additional detail in methods description and requests that the writing be revised. These have been all met with additional computational experiments presented by the authors and substantial edits.

I am in favor of publishing this with minor revisions.

We thank the reviewer for his valuable comments.

Reviewer #2 (Remarks to the Author):

I have read the manuscript and i found the answers to my questions acceptable. I have only one comment left. The authors state in the proteomics analysis

"As shown in Fig. 5C, the integration of four methods has the highest protein coverage 242 and can predict all 1001 proteins, while three of the four individual methods can predict a substantial lower number of proteins (250 for TRN, 547 for KEGG, 1,000 for PPI, 847 for 244 CPN). The prediction performance of the integrated method (0.55 ± 0.26) outperforms all 245 individual methods (0.41 ± 0.23 for TRN; 0.47 ± 0.23 for KEGG; 0.48 ± 0.26 for PPI; 0.52 ± 0.24 for CPN)".

I will argue that the difference between 1001 and 1000 (PPI) is not substantially different. Similarly the outperformance is not large (ie 0.55 ± 0.26 vs 0.52 ± 0.24 for CPN).

I think the authors should be more modest in their adjectives in this section.

We thank the reviewer for his comment and we are revising the section to reflect what we want to point out (PPI has similar coverage as in integration but lower PCC):

As shown in Fig. 5C, the integration of four methods shows higher coverage and higher performance when compared with the four individual methods. In terms of protein coverage, the integration can predict all 1,001 proteins, comparable only to the PPI method (1000 proteins), while the other three individual methods can predict a substantially lower number of proteins (250 for TRN, 547 for KEGG, 847 for CPN). However, the prediction performance of the PPI method (PCC: 0.48 ± 0.26 for PPI) is lower than using their integration (PCC: 0.55 ± 0.26). The integration method also outperforms the other two individual methods (PCC: 0.41 ± 0.23 for TRN; 0.47 ± 0.23 for KEGG) and is close to that of the CPN method (PCC: 0.52 ± 0.24), although the latter has a 15.4% less coverage. To directly compare sets of proteins that are covered in all methods, we focused their performance in the top 50 most variable proteins that are common among the five sets. Our results show that the integration (ensemble method) outperforms all other combinations with a PCC of 0.77 ± 0.27 , which is substantially better than all others (CPN, PCC= 0.69 ± 0.27 ; PPI, PCC= 0.60 ± 0.36 ; KEGG, PCC= 0.66 ± 0.30 , TRN, PCC= 0.37 ± 0.49). Interestingly, predicting first the target gene expression from the mRNA expression levels of the corresponding genes does not perform well, achieving a PCC of 0.34 ± 0.18 in the general case and PCC of 0.18 ± 0.51 for the 50 most variable proteins (Supplementary Fig. 23).